# Selection for Both Growth and Wood Properties in Chinese Fir Breeding Parents Based on a 6-Year Grafted Clone Test

Rong Huang [†], Runhui Wang [†], Ruping Wei, Shu Yan, Guandi Wu and Huiquan Zheng *

Guangdong Provincial Key Laboratory of Silviculture, Protection and Utilization, Guangdong Academy of Forestry, Guangzhou 510520, China; huangrong@sinogaf.cn (R.H.); wrh@sinogaf.cn (R.W.); weirp@sinogaf.cn (R.W.); yanshu@sinogaf.cn (S.Y.); wgd@sinogaf.cn (G.W.)

* Correspondence: zhenghq@sinogaf.cn; Tel.: +86-20-8758-4306; Fax: +86-20-8703-1245
† These authors contributed equally to this work.

**Abstract:** With the growing demand for high-quality timber, selection processes for both growth and wood properties are needed for multi-trait breeding programs in Chinese fir (*Cunninghamia lanceolata* (Lamb.) Hook.). The present study examined the variation and correlation of growth (tree height, diameter at breast height, stem volume, crown-width) and wood properties (wood basic density, hygroscopicity, and heart-wood ratio) traits of 201 Chinese fir breeding parents, aiming to select better parents for future multi-trait improvement. The results showed that significant differences ($p < 0.01$) regarding growth and wood property traits were observed among clones in an individual site and in a two-site joint analysis. The repeatability of the tested traits varied from 0.22 to 0.87. Strong and positive ($p < 0.01$) correlations were detected among the four growth traits, while wood basic density had a significant negative correlation ($p < 0.01$ or $0.05$) with the growth traits. A set of parent clones was shortlisted with substantial realized gains (ranging from 4.59% to 83.77%) in growth and wood traits. It was suggested that these selected parents could be used to improve the growth and wood quality of Chinese fir.

**Keywords:** *Cunninghamia lanceolata*; repeatability; correlation; realized gain

## 1. Introduction

The selection of the appropriate parents is a paramount step in tree breeding programs that will increase the chances of obtaining maximum genetic variability and generating superior recombinant progeny [1,2]. Accurate estimation of genetic parameters, especially breeding values, is of great significance for parent selection [3]. It is necessary to improve the precision of tree breeding values, thus improving the breeding efficiency. Several statistical methods have been developed and used for the prediction of breeding values under fixed, random, or mixed models, and using least squares, maximum likelihood, or Bayesian inference [4]. For forest trees in general, it is difficult to obtain accurate genetic parameters due to the effects of complex population structure, environmental factors, and unbalanced observed data [5–7]. The Best Linear Unbiased Prediction (BLUP) method was developed to dissect genetic and environmental effects, and was originally developed in dairy cattle breeding for the purpose of predicting breeding values. This method has become a flexible and powerful tool with which to achieve precise breeding value predictions in animal breeding programs [8–10]. Although the BLUP-biased method has been used for breeding programs in forestry since the 1980s [11], such as *Eucalyptus* [2,12,13], *Populus* [3,14,15], and *Acacia mearnsii* [16], it does not seem to be equally popular in plant breeding as it is in animals. The literature regarding coniferous tree breeding shows that little efforts have been made in the application of a BLUP-biased method for coniferous tree breeding [17–19].

Chinese fir (*Cunninghamia lanceolata* (Lamb.) Hook.) is an important coniferous tree species with great timber, afforestation, and ecological values, and occupies approximately

25% of plantations in China [20,21] and has also been historically introduced to other countries (e.g., New Zealand and Brazil) [22,23]. Chinese fir has received a considerable breeding effort with growth traits as the main breeding objective since the 1960s in China [20,24,25]. However, much less work has been done to improve wood properties in Chinese fir breeding programs [20]. This decade, the improvement of wood quality traits has been targeted as one of the primary breeding objectives because of increasing demands for high-quality timber [20]. To promote process of the multi-trait breeding programs, it is essential to select the most suitable parents for making crosses with the purpose of the improving both growth and wood properties. The prediction accuracy of genetic parameters (e.g., breeding values) is of great significance for the improvement of Chinese fir breeding efficiency regarding both growth and wood properties. Considering the advantages of the BLUP method, the method could be extremely useful for genetic parameter evaluations in Chinese fir breeding programs. However, as far as we know, few Chinese fir breeding programs have adopted the BLUP method to predict breeding values [26].

In this study, a total of 201 Chinese fir breeding parents were subjected to a 6-year grafted clone test that has proven to be a straightforward and effective strategy for parent selection [20,27]. The aim of the study was to analyze the variation of growth and wood property traits among the parent clones and the phenotypic correlations among the traits, and, finally, to select the best potential parent clones based on BLUP breeding values for future breeding programs.

## 2. Materials and Methods

### 2.1. Test Locations and Materials

The field works were carried out in two three-generation breeding gardens located at Longshan State Forest Farm (25°11′ N, 113°28′ E, alt. 286.5 m) and Xiaokeng State Forest Farm (24°42′ N, 113°48′ E, alt. 303 m), Guangdong, China. The annual mean temperature was 19.6 °C and 20.3 °C at Longshan State Forest Farm and Xiaokeng State Forest Farm, respectively, and the annual average rainfall was 1500 mm and 1530 mm in the two breeding gardens, respectively. Both of the gardens belonged to the yellowish red soil type. In total, 135 and 170 plus-trees (clones) were conserved at Longshan State Forest Farm and Xiaokeng State Forest Farm by using their scions grafted onto rootstocks, respectively. There was a total of 201 plus-trees represented across the two gardens. The clones were assigned to the field by a randomized distribution method based on Microsoft Excel's randomized arrangement program, with 5 ramets for each clone at Longshan State Forest Farm and 5 to 10 ramets at Xiaokeng State Forest Farm. Ramets were planted with a spacing of 3 m × 3 m at Longshan State Forest Farm and 4 m × 4 m at Xiaokeng State Forest Farm.

### 2.2. Phenotypic Measurements

The traits of total tree height (H), diameter at breast height (DBH, 1.3 m), and crown width (CW) were measured for each tree at 6 years of age. The tree stem volume (V) was calculated on the basis of H and DBH according to the following equation [28]:

$$V = 5.8777042 \times 10^{-5} \times DBH^{1.9699831} \times H^{0.89646157}$$

For each tree, three ramets of similar sizes were randomly selected for the assessment of wood basic density (WBD), hygroscopicity (Hy), and heart/wood ratio (HR) with the help of an increment core method that was described by Zheng et al. in detail [20,24]. Specifically, a 5.02 mm diameter bark-to-pith core of wood was extracted at breast height from each tree with an increment borer, and then immediately put into a plastic tube with two sealed ends.

Herein, WBD (g/cm$^3$) was determined using a water displacement method with two weights for each tree, and then evaluated according to the following formula [29,30]:

$$WBD = 1/((W_1 - W_2)/W_2 + 1/\rho_{CW})$$

where $W_1$ and $W_2$ represent the water-saturated weight and oven dry weight, respectively, and $\rho_{CW}$ represents the wood cell wall component density, which was around 1.53 g/cm$^3$.

The hygroscopicity (Hy) was calculated as:

$$Hy = (W_1 - W_2)/W_1$$

where $W_1$ and $W_2$ are consistent with the aforementioned weights.

The heart/wood ratio (HR) was calculated theoretically based the value of ($r^2 \times \Pi)/(R^2 \times \Pi)$, where r and R represent the radii of heartwood and whole wood, respectively [24].

### 2.3. Statistical Analysis

All collected data were entered into Microsoft Excel 2019 and analyzed by one-way analysis of variance (ANOVA) using statistical package R (v.3.6.0) to determine whether the traits were significantly different among the clones. Linear mixed models were used for separate analyses of individual sites and for two-site interaction analyses:

$$\text{Individual site statistical model: } y_{ij} = u + \text{clone}_i + e_{ij}$$

Two-site joint analysis statistical model:

$$y_{ij} = u + \text{site}_i + \text{clone}_j + \text{site}_i * \text{clone}_j + e_{ij}$$

where $y_{ij}$ and $u$ represent the phenotypic observation value for the $j$ clone of $i$ parent and the mean value of the phenotypic observation; $\text{clone}_i$ and $e_{ij}$ are the random effects of the $i$ parent and the random error of the j clone of the $i$ parent. $\text{site}_i$ represents the site effect, and $\text{site}_i * \text{clone}_j$ represents the interaction effect of the site and the clone.

In addition, the phenotypic and genotypic coefficients of variation for each trait were also calculated by following the method of Burton using R-software (v.3.6.0) [26]. The phenotypic and genetic coefficients of correlation between two traits were evaluated by correlation analysis using R-software, respectively. Clonal repeatability (R) for each trait was estimated as follows [31]:

$$R = \frac{\delta_b^2}{\delta_b^2 + \frac{\delta_w^2}{K}}$$

where $\delta_b^2$ is the variance component between clones, $\delta_w^2$ is the residual variance component, and $K$ is the ramet number of a certain clone.

Breeding values for three valuable traits, including tree height, diameter at breast height, and tree stem volume, were also calculated based on BLUP model equation [32] in ASReml version 3.0.

Breeding values were given normalized treatment, as follows:

$$y = \frac{x - x_{min}}{x_{max} - x_{min}}$$

where $x$, $x_{max}$, and $x_{min}$ represent the observed, maximum, and minimum values of a certain breeding value. Then, the normalized breeding values of the tested traits' clone ranking were plotted to reveal whether the clones had a consistent ranking across the traits.

Realized gain ($G$) was calculated using the formula:

$$G = \left(\overline{X}i - \overline{X}\right)/\overline{X} \times 100\%$$

where $\overline{X}i$ and $\overline{X}$ are the $i$ parent clone and overall trait mean, respectively.

The package "ggplot2" [33] in R was used for the figures plot.

## 3. Results and Discussion

### 3.1. Differences among Parent Clones in Growth and Wood Property Traits

From a wood utilization perspective, it is of great importance for breeders to know the variation and heritability of vital growth and wood traits to improve their realized gain [34,35]. An analysis of variance among the tested Chinese fir breeding parents was conducted for the seven quantitative traits (tree height (H), diameter at breast height (DBH), crown width (CW), tree stem volume (V), wood basic density (WBD), hygroscopicity (Hy), and heart/wood ratio (HR). In addition, different genetic parameters (i.e., phenotypic and genotypic coefficients of variation (PCV and GCV) and repeatability (R)) were also calculated to determine the variation in the growth and wood property traits of the breeding parents.

In terms of growth property traits, the averages of H, DBH, V, and CW were 7.1 and 5.2 m, 13.2 and 12.2 cm, 0.0609 and 0.0346 $m^3$, and 3.5 and 3.3 m in Longshan and Xiaokeng, respectively (Table 1). For the wood property traits, the averages of WBD, Hy, and HR were 0.307 and 0.293 $g/cm^3$, 265.01% and 275.94%, and 34.99% and 19.48% in Longshan and Xiaokeng, respectively (Table 1). When integrating the two tested sites, the averages of H, DBH, V, CW, WBD, Hy, and HR were 6.2 m, 12.82 cm, 0.0522 $m^3$, 3.38 m, 0.300 $g/cm^3$, 270.69%, and 26.94%, respectively. The ANOVA results revealed that all seven growth and wood property traits differed significantly ($p < 0.01$) among the Chinese fir breeding parent clones (Table 1). Except for HR, site-by-clone interactions were significant ($p < 0.01$) for the remanent six traits (Tables 1 and S1). Ontogenetic plasticity may account for the significant genotype-by-environment interaction in these traits [36]. Phenotypic and genotypic coefficients of variation (PCV and GCV) are two important parameters reflecting the degree of variation that exists in a given breeding population [37,38]. The estimations of the two parameters (PCV and GCV) for the studied traits are presented in Table 1. Among all of the traits, stem volume (V) displayed the most substantial variation, with PCV and GCV values of 60.17% and 49.81% in Longshan and 75.02% and 52.43% in Xiaokeng, as well as means of 68.65% and 48.77%, followed by the heart/wood ratio (HR) (Longshan: PCV = 46.50%, GCV = 12.65%; Xiaokeng: PCV = 72.87%, GCV = 39.16%; mean: PCV = 63.40%, GCV = 26.28%). The traits of Hy, CW, DBH, and H showed lower PCV and GCV values compared to V and HR in Longshan and Xiaokeng (Table 1). On the contrary, wood basic density (WBD) showed the lowest values (Longshan: PCV = 12.10%, GCV = 6.72%; Xiaokeng: PCV = 12.45%, GCV = 7.04%; mean: PCV = 12.38%, GCV = 6.33%). Zheng et al. [20] reported that the PCV values of the H, DBH, V, and CW traits were 13.3%, 11.9%, 30.6%, and 13.3% in the 6-year-old Chinese fir breeding parents. In this study, the results showed higher estimated PCV values for these traits (H: 20.79%–25.10%, DBH: 23.26%–28.99%, V: 60.17%–75.02%, CW: 20.93%–28.60%), indicating the presence of substantial variation among the breeding parents in the present study. In general, the GCV had a lower value than PCV for all the traits in this study. The differences between GCV and PCV were smaller for the four growth traits (H, DBH, V, and CW) than for the three wood traits (WBD, Hy, and HR) (Table 1). This indicates that the juvenile wood traits are more susceptible to environmental conditions relative to growth traits [39], of which we should be aware during early selection for wood property traits in Chinese fir trees.

Repeatability is a significant parameter for parent selection with the help of clone tests, as it is an index that reflects a trait's stability [40]. Higher repeatability means that the trait is less affected by the external environment [41]. Our results showed that high repeatability was found for the growth property traits (H, DBH, V, and CW), with a range of 0.63 to 0.87, and moderate repeatability was observed for the wood property traits (WBD, Hy, and HR) (Table 1). These results further suggest that the majority of the variability of growth and wood property traits was strongly controlled by heredity, which is in agreement with the finding observed by Zheng et al. [20], who reported moderate to high repeatability (from 0.53 to 0.70) in these traits. Therefore, it was concluded that phenotypic value could be used for the selection of these traits.

**Table 1.** Descriptive statistics of growth and wood property traits of breeding parent clones of Chinese fir.

| Site | Parameter | H (m) | DBH (cm) | V (m$^3$) | CW (m) | WBD (g/cm$^3$) | Hy (%) | HR (%) |
|---|---|---|---|---|---|---|---|---|
| Longshan | Mean ± SD | 7.1 ± 1.5 | 13.2 ± 3.1 | 0.0609 ± 0.0366 | 3.5 ± 0.7 | 0.307 ± 0.037 | 265.01 ± 39.80 | 34.99 ± 16.26 |
| | Range | 2.4–13.0 | 5.2–22.5 | 0.0042–0.2701 | 1.4–6.0 | 0.211–0.398 | 185.31–408.57 | 3.39–90.40 |
| | *F*-statistic | 5.39 ** | 7.68 ** | 6.64 ** | 3.05 ** | 2.10 ** | 2.31 ** | 1.28 ** |
| | PCV (%) | 20.79 | 23.26 | 60.17 | 20.93 | 12.10 | 15.01 | 46.50 |
| | GCV (%) | 16.43 | 19.84 | 49.81 | 14.14 | 6.72 | 8.82 | 12.65 |
| | R | 0.81 | 0.87 | 0.85 | 0.67 | 0.52 | 0.57 | 0.22 |
| Xiaokeng | Mean ± SD | 5.2 ± 1.3 | 12.2 ± 3.6 | 0.0346 ± 0.0331 | 3.3 ± 0.9 | 0.293 ± 0.037 | 275.94 ± 41.66 | 19.48 ± 14.2 |
| | Range | 2.4–8.7 | 4.1–25.5 | 0.0031–0.2281 | 1.4–8.0 | 0.214–0.398 | 185.27–401.45 | 0.82–90.45 |
| | *F*-statistic | 3.81 ** | 7.20 ** | 6.12 ** | 3.05 ** | 1.98 ** | 2.12 ** | 1.84 ** |
| | PCV (%) | 25.10 | 28.99 | 75.02 | 28.60 | 12.45 | 15.10 | 72.87 |
| | GCV (%) | 18.90 | 25.33 | 52.43 | 18.56 | 7.04 | 8.83 | 39.16 |
| | R | 0.74 | 0.86 | 0.84 | 0.63 | 0.49 | 0.53 | 0.46 |
| Joint analyses | Mean ± SD | 6.2 ± 1.7 | 12.82 ± 3.38 | 0.0522 ± 0.0358 | 3.38 ± 0.85 | 0.300 ± 0.04 | 270.69 ± 41.12 | 26.94 ± 17.08 |
| | Range | 2.4–13.0 | 4.1–25.5 | 0.0031–0.2701 | 1.4–8.0 | 0.211–0.398 | 185.27–408.57 | 0.82–90.45 |
| | PCV (%) | 27.21 | 26.36 | 68.65 | 25.28 | 12.38 | 15.19 | 63.40 |
| | GCV (%) | 14.83 | 21.56 | 48.77 | 15.34 | 6.33 | 7.92 | 26.28 |
| | R | 0.34 | 0.56 | 0.44 | 0.27 | 0.20 | 0.19 | 0.12 |
| | *F*-statistic (clone) | 5.33 ** | 9.49 ** | 7.64 ** | 3.37 ** | 2.27 ** | 2.44 ** | 1.65 ** |
| | *F*-statistic (site × clone) | 2.56 ** | 2.62 ** | 2.97 ** | 1.74 ** | 1.42 ** | 1.65 ** | 1.04 |

Note: H: total tree height, DBH: diameter at breast height, V: stem volume, CW: crown width, WBD: wood basic density, Hy: hygroscopicity, HR: heart/wood ratio, SD: standard deviation, PCV: phenotypic coefficient of variation, GCV: genetic coefficient of variation, R: repeatability, **: significant difference at the 0.01 level.

The above findings imply that the tested traits vary extensively and significantly among Chinese fir breeding parents, providing substantial variation for potential parent selection and improvement of these growth and wood property traits.

*3.2. Genetic and Phenotypic Correlations among Traits*

Knowledge of the correlation between traits is of utmost importance in the multi-trait breeding process, and can provide valuable information for comprehensive utilization of the available variation in target traits [40]. To take wood property traits into account in Chinese fir breeding programs, breeders should acquire information on the magnitude of correlation between wood and growth characteristics. Here, an analysis of genetic and phenotypic correlation between the wood and growth traits was conducted.

In the present study, significantly and highly positive genetic and phenotypic correlations were observed between DBH and V (genetic correlation: r = 0.73–0.97, *p* < 0.01; phenotypic correlation: r = 0.94–0.95, *p* < 0.01), H and V (genetic correlation: r = 0.84, *p* < 0.01; phenotypic correlation: r = 0.77–0.83, *p* < 0.01), and H and DBH (genetic correlation: r = 0.76–0.77, *p* < 0.01; phenotypic correlation: r = 0.65–0.73, *p* < 0.01) in the two sites (Table 2). These traits (DBH, H, and V) also had significantly positive associations with CW (genetic correlation: r = 0.47–0.81, *p* < 0.01; phenotypic correlation: r = 0.35–0.60, *p* < 0.01). A positive correlation was also found in these traits when performing the two-site jointing analyses (genetic correlation: r = 0.61–0.98, *p* < 0.01; phenotypic correlation: r = 0.38–0.93, *p* < 0.01) (Table 2). The positive associations between these growth traits agreed with previous studies reported by Zheng et al. [20,24] and Wang et al. [26]. The positive association between these growth traits suggests that they are not inherited independently.

A synthesis selection process for rapid growth with high wood quality may contribute to the improvement of the wood yield and quality of Chinese fir [42]. Wood basic density (WBD) is one of the most crucial wood properties influencing the quality of timber [43,44]. In this study, the correlation estimate results showed that WBD presented a significantly negative correlation with the four growth traits (i.e., H, DBH, V, and CW) (genetic correlation: r = −0.40−−0.25, *p* < 0.01; phenotypic correlation: r = −0.27−−0.13, *p* < 0.01 or 0.05) in the two sites (Table 2). The negative correlation between wood and growth traits in Chinese fir has also been reported by Hu et al. [45,46], Li et al. [47], and Zheng et al. [20]. This indicates that the growth rate may have a great effect on the wood basic density. Hygroscopicity (Hy) is another wood trait that has a great impact on the durability,

dimensional stability, and hydrophobicity of timber [48,49]. Here, we found that Hy has less association with the growth traits, but a significantly and strongly negative correction (r = −0.99, *p* < 0.01) with WBD (Table 2). Previous studies have demonstrated that an increase in WBD may result in a reduction in the Hy [50–52], which was also confirmed in the present study. To this end, selection for faster-growing features might result in a reduction in wood quality, of which we need to be aware in Chinese fir breeding programs.

**Table 2.** Estimates of genetic (above the diagonal) and phenotypic (below the diagonal) correlations among seven traits of Chinese fir.

| Site | Trait | H | DBH | V | CW | WBD | Hy | HR |
|------|-------|------|------|------|------|------|------|------|
| Longshan | H | 1.00 | 0.76 ** | 0.84 ** | 0.47 ** | −0.27 ** | 0.26 ** | 0.10 |
| | DBH | 0.73 ** | 1.00 | 0.73 ** | 0.73 ** | −0.40 ** | 0.40 ** | 0.20 ** |
| | V | 0.83 ** | 0.95 ** | 1.00 | 0.68 ** | −0.37 ** | 0.37 ** | 0.20 ** |
| | CW | 0.36 ** | 0.59 ** | 0.56 ** | 1.00 | −0.25 ** | 0.22 ** | 0.16 ** |
| | WBD | −0.17 ** | −0.27 ** | −0.25 ** | −0.13 * | 1.00 | −0.99 ** | 0.41 ** |
| | Hy | 0.16 ** | 0.27 ** | 0.24 ** | 0.11 | −0.99 ** | 1.00 | −0.40 ** |
| | HR | 0.04 | 0.07 | 0.06 | 0.03 | 0.19 ** | −0.18 ** | 1.00 |
| Xiaokeng | H | 1.00 | 0.77 ** | 0.84 ** | 0.66 ** | −0.26 ** | 0.24 ** | 0.26 ** |
| | DBH | 0.65 ** | 1.00 | 0.97 ** | 0.81 ** | −0.34 ** | 0.32 ** | 0.24 ** |
| | V | 0.77 ** | 0.94 ** | 1.00 | 0.80 ** | −0.31 ** | 0.29 ** | 0.21 ** |
| | CW | 0.35 ** | 0.60 ** | 0.54 ** | 1.00 | −0.30 ** | 0.26 ** | −0.02 |
| | WBD | −0.14 * | −0.22 ** | −0.21 ** | −0.13 * | 1.00 | −0.99 ** | −0.20 ** |
| | Hy | 0.12 * | 0.21 ** | 0.19 ** | 0.13 * | −0.99 ** | 1.00 | 0.20 ** |
| | HR | 0.08 | 0.17 ** | 0.13 | 0.10 | 0.14 * | −0.13 * | 1.00 |
| Joint analyses | H | 1.00 | 0.83 ** | 0.86 ** | 0.61 ** | −0.20 ** | 0.19 ** | 0.21 ** |
| | DBH | 0.62 ** | 1.00 | 0.98 ** | 0.79 ** | −0.41 ** | 0.39 ** | 0.17 ** |
| | V | 0.78 ** | 0.93 ** | 1.00 | 0.75 ** | −0.36 ** | 0.35 ** | 0.14 * |
| | CW | 0.38 ** | 0.60 ** | 0.56 ** | 1.00 | −0.26 ** | 0.23 ** | 0.05 |
| | WBD | −0.06 | −0.23 ** | −0.19 ** | −0.10 | 1.00 | −0.99 ** | −0.04 |
| | Hy | 0.04 | 0.22 ** | 0.18 ** | 0.09 | −0.99 ** | 1.00 | 0.07 |
| | HR | 0.30 ** | 0.15 * | 0.19 ** | 0.14 ** | 0.20 ** | −0.20 ** | 1.00 |

Note: H, total tree height; DBH, diameter at breast height; V, stem volume; CW, crown width; WBD, wood basic density; Hy, hygroscopicity; HR, heart/wood ratio. Single asterisks (*) and two asterisks (**) indicate significant differences at the 0.05 and 0.01 levels, respectively.

*3.3. Parent Clone Selection*

With the increasing concerns regarding wood quality traits in Chinese fir, the parent selection process, on the basis of a combination of growth and wood property traits, should be taken into account in a long-term Chinese breeding program [20]. Total tree height, diameter at breast height, and stem volume are three important growth properties, and wood basic density (WBD) is always considered as a critical wood characteristic influencing the end-product quality [20,43,44]. These traits are mainly governed by heredity and are less affected by the environment factors, which is evidenced by our results regarding repeatability (Table 2). Thus, the four traits, as well as the heart/wood ratio, were used as the parent selection criteria for the purpose of obtaining fast-growing parents with high-quality wood traits in the current study.

BLUP breeding values have been considered as superior parameters for use in parent selection strategies [3,17–19]. The breeding values regarding the H, DBH, V, and WBD of the parent clones were estimated using the BLUP method, as shown in Tables S2 and S3. To rank the order of parent breeding values, they were subjected normalized treatment. The rank order of the normalized breeding values of H, DBH, V, and WBD for the Chinese fir breeding parents are shown in Figure 1. Most of the parent clones had a consistent ranking across the four traits (Figure 1). Among all the parents, 20 and 16 parents harbored breeding values over 0 for H, DBH, V, and WBD in Longshan and Xiaokeng, respectively (Tables S2 and S3). When only considering growth property traits (H, DBH, and V), 51 and 59 parents had breeding values over 0 in the two sites, respectively (Tables S2 and S3). These may potentially be valuable breeding parents for selection.

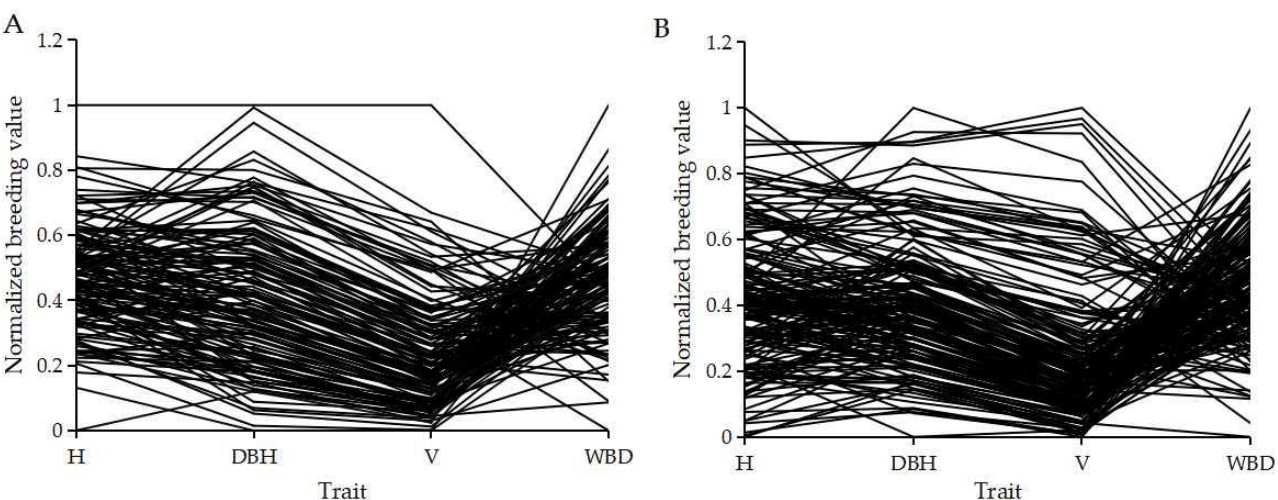

**Figure 1.** Rankings for normalized breeding values for four traits in Chinese fir breeding parents ((**A**): Longshan, (**B**): Xiaokeng). H: total tree height, DBH: diameter at breast height, V: stem volume, WBD: wood basic density. Each black line represents one breeding parent.

The ultimate goal of this study was to address the selection of excellent Chinese fir parents, both in growth and wood properties, for multi-trait breeding programs. So far, much less work has been conducted aiming for the improvement of wood property traits in the multi-trait selection of Chinese fir trees [20,24]. Therefore, selection for growth and wood property traits was performed using a Venn diagram method with breeding values over 0, and WBD or HR exceeded the overall means as selection criteria. With the help of the criteria, the selected clones could be regarded as potential breeding parents that were excellent both in their growth and wood properties. On the basis of breeding values over 0 for growth property traits (H, DBH, and V), together with WBD exceeding the overall mean thresholds, a total of 21 and 10 parent clones were selected with selection intensities of 1.554 and 2.023 in Longshan and Xiaokeng, respectively (Figure 2A,C). In this case, the selected parents had higher mean DBH (Longshan: 15.5 cm; Xiaokeng: 12.5 cm) and WBD (Longshan: 0.320 g/cm$^3$; Xiaokeng: 0.300 g/cm$^3$) values than the overall means (Longshan: DBH = 13.2 cm, WBD = 0.307 g/cm$^3$; Xiaokeng: DBH = 12.2 cm, WBD = 0.293 g/cm$^3$) (Table 3). The mean breeding values for H, DBH, V, and WBD of the selected parent clones were much higher than those of all parent clones (Figure 3). They exhibited high realized gains (*G*) in stem volume (43.45% for Longshan and 79.26% for Xiaokeng), and moderate levels in DBH (*G* = 18.04% and 29.15%) and H (*G* = 13.31% and 24.64%); however, they tended to have relative lower realized gains in WBD (*G* = 4.59% and 8.31%) (Tables 3, S4 and S5). When considering the heart/wood ratio (HR) instead of WBD, a set including 29 and 30 parent clones with large DBH, H, V, and HR values were shortlisted under selection intensities of 1.372 and 1.489 in Longshan and Xiaokeng (Figure 2B,D), respectively. The selected parent clones showed much higher mean breeding values for the traits in comparison with the means of all parent clones (Figure 4). More importantly, substantial realized gains in V, HR, DBH, and H were achieved, ranging from 15.00% to 83.77% (Tables 3, S6 and S7). The results suggest that these selected parent clones may be the best choice for multi-trait breeding programs.

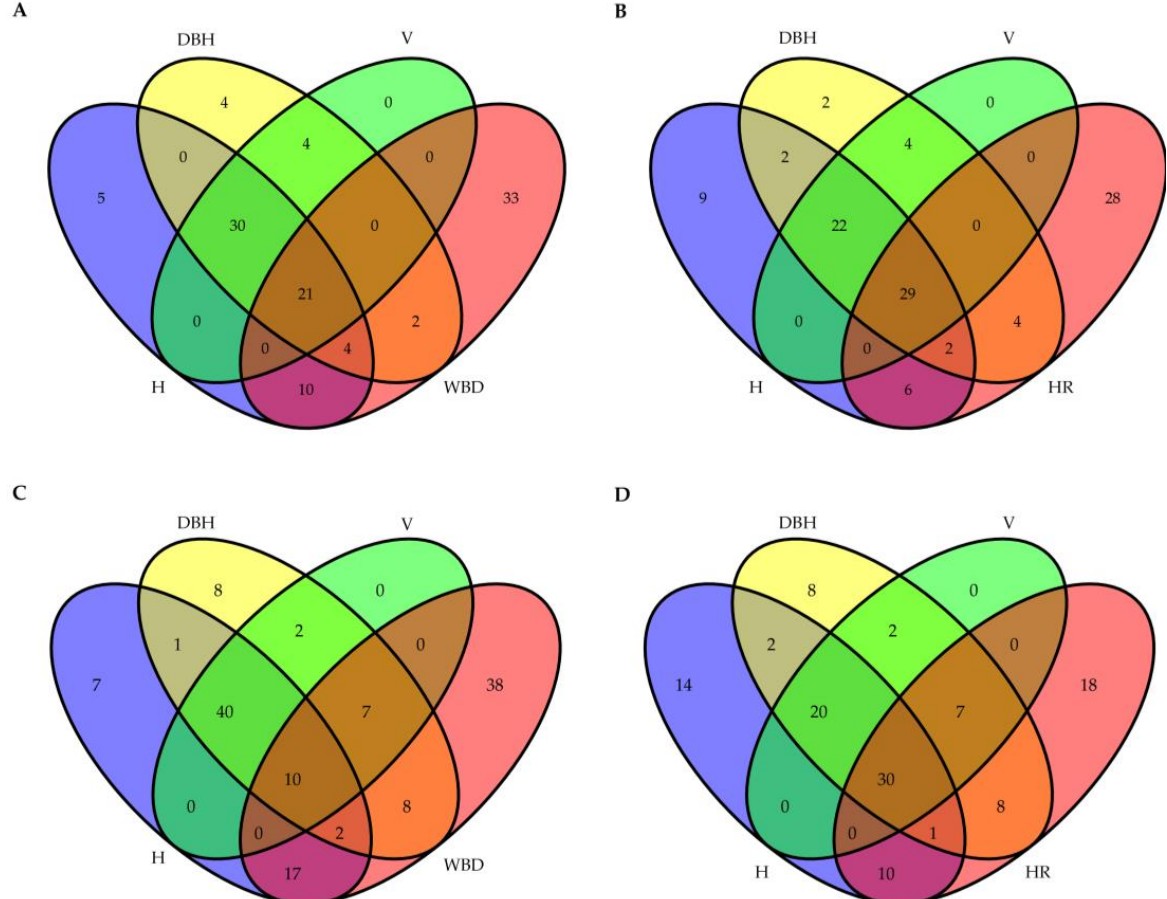

**Figure 2.** The four-set Venn diagrams of growth and wood property traits with breeding values over thresholds of 0, illustrating the number of overlapping and non-overlapping breeding parents of Chinese fir trees in Longshan (**A**,**B**) and Xiaokeng (**C**,**D**). H: total tree height, DBH: diameter at breast height, V: stem volume, WBD: wood basic density, HR: heart/wood ratio.

**Table 3.** Overall means for H, DBH, V, WBD, and HR under different selection criteria, used for the selection of Chinese fir breeding parents to realize genetic gain.

| Site | Trait | Selection for H, DBH, V, and WBD | | Selection for H, DBH, V, and HR | |
|---|---|---|---|---|---|
| | | Mean ± SD | *G* (%) | Mean ± SD | *G* (%) |
| Longshan | H (m) | 8.0 ± 0.7 | 13.31 ± 9.61 | 8.19 ± 0.98 | 15.00 ± 13.80 |
| | DBH (cm) | 15.5 ± 1.7 | 18.04 ± 12.97 | 15.89 ± 1.96 | 20.80 ± 14.87 |
| | V (m$^3$) | 0.0876 ± 0.0210 | 43.45 ± 34.91 | 0.0938 ± 0.0296 | 52.82 ± 49.89 |
| | WBD (g/cm$^3$) | 0.320 ± 0.011 | 4.59 ± 3.18 | - | - |
| | HR (%) | - | - | 42.42 ± 6.88 | 21.22 ± 19.67 |
| Xiaokeng | H (m) | 5.3 ± 0.7 | 24.64 ± 11.04 | 5.28 ± 0.73 | 25.06 ± 13.74 |
| | DBH (cm) | 12.5 ± 2.2 | 29.15 ± 15.09 | 12.52 ± 2.44 | 31.23 ± 19.50 |
| | V (m$^3$) | 0.0400 ± 0.0273 | 79.26 ± 55.73 | 0.0400 ± 0.0253 | 83.77 ± 56.21 |
| | WBD (g/cm$^3$) | 0.300 ± 0.023 | 8.31 ± 6.71 | - | - |
| | HR (%) | - | - | 19.49 ± 6.77 | 45.58 ± 34.73 |

Note: H: total tree height, DBH: diameter at breast height, V: stem volume, WBD: wood basic density, HR: heart/wood ratio, *G*: realized gain.

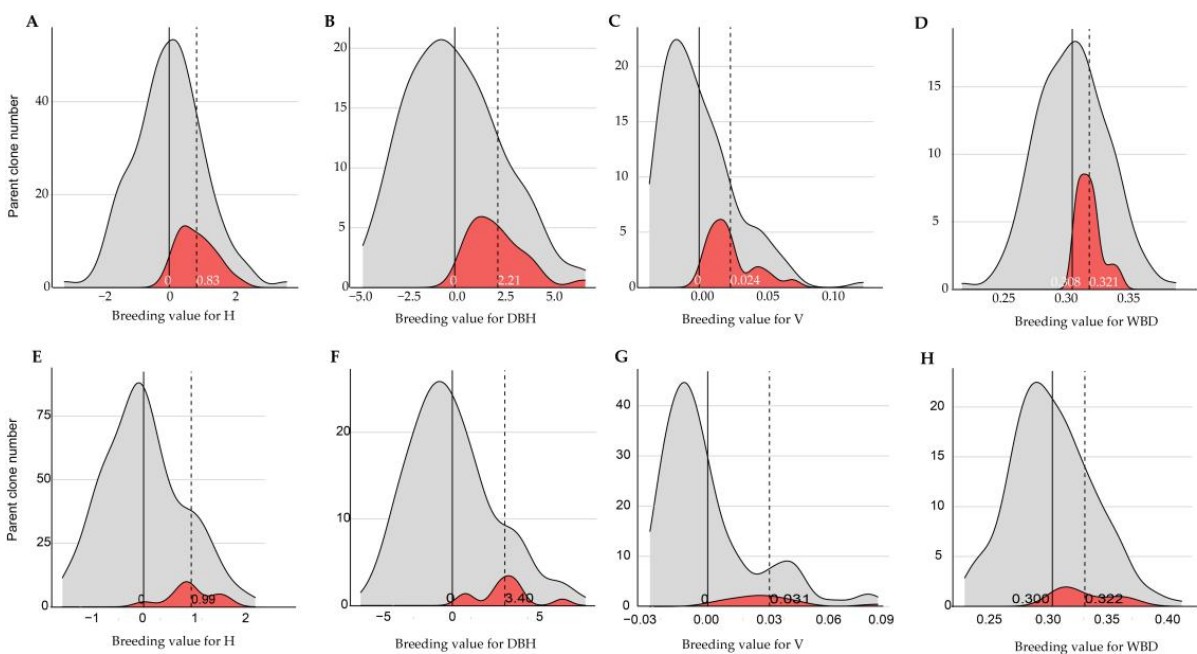

**Figure 3.** Number distribution of selected parent clones of Chinese fir trees under selection intensities of 1.554 and 2.023 based on breeding value in Longshan (**A–D**) and Xiaokeng (**E–H**). H: total tree height, DBH: diameter at breast height, V: stem volume, WBD: wood basic density, *G*: realized gain. Grey indicates all parents; red represents the selected parents with excellent growth and wood traits; solid line indicates the mean breeding values of all parents; dotted line represents the mean breeding values of the selected parents with excellent growth and wood traits.

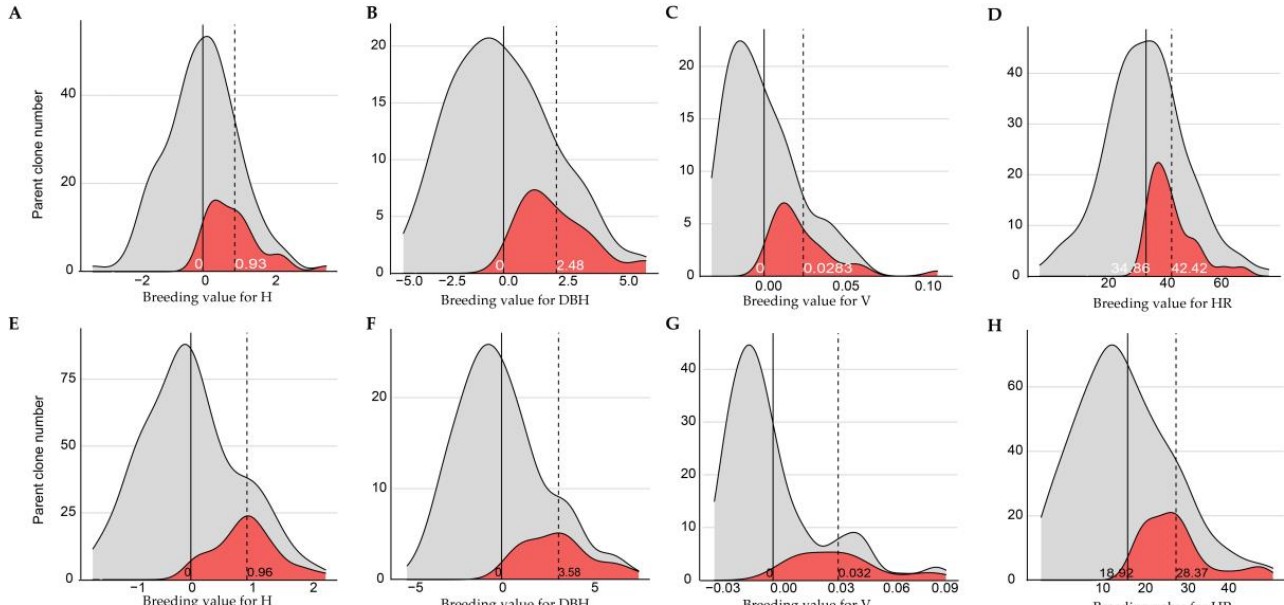

**Figure 4.** Number distribution of selected parent clones of Chinese fir under selection intensities of 1.372 and 1.489, based on breeding value in Longshan (**A–D**) and Xiaokeng (**E–H**). H: total tree height, DBH: diameter at breast height, V: stem volume, WBD: wood basic density, *G*: realized gain. Grey indicates all parents; red represents the selected parents with excellent growth and wood traits; solid line indicates the mean breeding values of all parents; dotted line represents the mean breeding values of the selected parents with excellent growth and wood traits.

## 4. Conclusions

In the current study, a total of 201 Chinese fir breeding parents were used for the assessment of growth and wood property traits in a 6-year grafted clone-testing process. Our results revealed that substantial variability in growth and wood property traits could be found among the breeding parents. A significant and negative association was observed between wood traits (WBD) and growth traits (i.e., H, DBH, V, CW). Using both growth and wood property traits as our selection objectives, parent clones with large DBH, H, V, WBD, and HR values were shortlisted, which yielded substantial realized gains. It was suggested that the selected parents were excellent in terms of the growth and wood property traits that could improve the growth and wood quality of Chinese fir.

**Supplementary Materials:** The following supporting information can be downloaded at: https://www.mdpi.com/article/10.3390/f14091774/s1, Table S1: Analyses of variance for seven traits of Chinese fir from two test sites; Table S2: Breeding values of H, DBH, V, and WBD for the Chinese fir breeding parents in Longshan; Table S3: Breeding values of H, DBH, V, and WBD for the Chinese fir breeding parents in Xiaokeng; Table S4: Responses in H, DBH, V, and WBD for the selection of Chinese fir breeding parents in Longshan; Table S5: Responses in H, DBH, V, and WBD for the selection of Chinese fir breeding parents in Xiaokeng; Table S6: Responses in H, DBH, V, and HR for the selection of Chinese fir breeding parents in Longshan; Table S7: Responses in H, DBH, V, and HR for the selection of Chinese fir breeding parents in Xiaokeng.

**Author Contributions:** Software, R.W. (Runhui Wang); formal analysis, R.W. (Runhui Wang); investigation, R.W. (Runhui Wang), H.Z., R.W. (Ruping Wei), S.Y. and G.W.; writing—original draft preparation, R.H. and R.W. (Runhui Wang); writing—review and editing, R.H. and H.Z.; visualization, R.W. (Runhui Wang); supervision, H.Z.; funding acquisition, H.Z. All authors have read and agreed to the published version of the manuscript.

**Funding:** This research was funded by the Key-Area Research and Development Program of Guangdong Province (No. 2020B020215001) and the National Key Research and Development Plan Project Sub-Subject (No. 2022YFD2200201-6).

**Data Availability Statement:** Data are available for research upon request.

**Conflicts of Interest:** The authors declare no conflict of interest.

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
