# Peer review of "Selection for Both Growth and Wood Properties in Chinese Fir Breeding Parents Based on a 6-Year Grafted Clone Test"

_forests, doi:10.3390/f14091774_

Round 1

Reviewer 1 Report

   This manuscript investigates the genetic relationships between growth traits and wood properties and the possibility of selection by multi-trait for the clonal test in 6-years Chinese far. It will be important findings for breeding strategies in this species.

   However, there are some questions and sentences that need correction, so please check and correct them.

   First, in this study, each trait was evaluated in the 6th year, but the 6th year corresponds to juvenile wood in many Gymnosperms, it's wood properties are generally different from mature wood.  Therefore, it is considered that the evaluation in the 6th year may not properly evaluate the true performance of the clones. You need to explain that the 6th year is juvenile wood and that the properties may differ from those of mature wood. Also, if there are any references that reports a comparison of juvenile and mature wood in terms of density and heart-wood ratio in this tree species, please introduce it in the text.

Also, on lines 155-156, you explain that the reason why the difference between PCV and GCV was greater for wood traits than for growth traits is that wood traits are more susceptible to environmental conditions. However, this explanation does not seem to be general. Many papers have shown that wood traits such as Young's modulus and density are often less affected by the environment than growth traits. Therefore, the results in this study could be due to the evaluation of juvenile wood whose performance of clone is not stable. It might be also the same reason why the repeatability of wood traits was lower than that of growth traits.

Please reconsider the flow from the intro to the discussion.

Next, when predicting BLUP, etc., you are analyzing each test site separately, but since the some clones are planted in two test sites, it is better to integrate the two test sites and analyze 201 clones together. You can analyze with a linear mixed model with the test site as the fixed effect, the clone effect and the interaction between the clone and the test site as the random effect. All figures and tables will also show only the combined results of the two sites.

Other points that require minor corrections are:

L90   I think the explanation of the calculation formula for basic density is not sufficient. I didn't understand what 0.346 meant, so please describe it.

L106   I don't think the calculation method of repeatability is a common one. I think that the general method is to calculate using genetic and phenotypic variance instead of calculating from the F value.

L200-201   Is “strongly positive correction” a mistake for “strongly negative correction”? Please confirm.

Figure 2   Please describe in the caption that the numbers in the Venn diagram are the number of breeding parents with breeding values over0 for each trait.

Author Response

Dear reviewer,

Thank you very much for very helpful and constructive comments. We received the reports from the editor on 8 August, 2023, and then revised the manuscript according to the comments of reviewers. All changes to the original manuscript were marked using the track-changes feature of Microsoft Word, which were also highlighted in red. We are now re-submitting our manuscript that has been significantly revised according to these comments and suggestions. Please let us know if additional changes are needed. The point-by-point response has been uploaded, please see the attachment.

Sincerely,

Huiquan Zheng

Reviewer 2 Report

The work is interesting. I found a few typos (e.g. line 277 "n" -> "in" ). It is not revealing as far as methodological aspects are concerned, but it has value as a source publication documenting the progress of breeding work.

I have a few comments that could improve the work:

The introduction lacks a more general perspective on the methodology of breeding programmes in forestry. Where they are conducted (e.g. USA, Sweden, Finland, New Zealand, Australia) the programs are based on BLUP as evidenced by publications from 50 years ago. In principle, it is a kitchen about which not much is written. On the other hand, the practical implementation of these methods has not taken place in many countries for many reasons. It is worth adding this thread to the discussion. It seems that the authors are not familiar with the publication: White, T. L., & Hodge, G. R. (1989). Predicting breeding values with applications in forest tree improvement (Vol. 33). Springer Science & Business Media.

- Comments on the methodology:

1. that the fir tree was propagated by grafting we learn from the title and the conclusion. There is nothing about this in the methodology. 

2. how the clones were distributed "randomly" is too general a statement - (line 73)

3. referring to Burton's [26] simple method as an assessment of repeatability in my opinion is not appropriate. Firstly, this is not a source publication but a self-citation. Secondly, in the case of clones, heritability in the strict sense can easily be calculated. (Falconer, D. S. (1996). Introduction to quantitative genetics. Pearson Education India.)

4 Is three trees in a clone sufficient repetition?  ( line 82) How does the standard error not stated relate to this. Please address this in the discussion.

5. figure 1 is unclear. In fact, it contributes almost nothing. 

6. The genotype x environment interaction has not been calculated. This may be the case but requires comment in the discussion.

7. It is not clear what type of correlation is used (tab 2). What about genetic correlations?

8. what does the abbreviation SD mean in the tables? (Standard deviation? )

9. what was the intensity of selection (figs 3 and 4)? A gain was given but how was this calculated?

10) What does very "excelent both in growth and wood properties" mean? Is this just a general statement or is there an objective criterion. If so, what is it?

The paper is full of such generalisations and understatements probably obvious to the authors but unclear to the outside reader. These should be clarified before publication

In my opinion it would be better if the discussion was separated from the results. It is residual in this form and, in my opinion, needs to be developed. 

I have pointed out methodological shortcomings here, which do not invalidate the essence of the work. The results will not be fundamentally changed either, but will be given in a different light. I understand that it is not easy to present results for so many genotypes and what has worked out quite well anyway.

I found a few typos (e.g. line 277). But overall the work is comprehensible and reads well

Author Response

(The authors gave the same response as above.)

Round 2

Reviewer 1 Report

Thank you for your diligent efforts to improve and revise the manuscript in response to my observations. I consider this paper to have reached publication quality.

Reviewer 2 Report

II understand the approach of combining results and discussion. I am glad that my comments have helped to improve it. 

In my opinion, fig 1. is simply unreadable. You cannot see the ranking of the clones there because there are too many. The data are also presented in the supplement.